# PLUGIn: A simple algorithm for inverting generative models with recovery guarantees

**Babhru Joshi**    **Xiaowei Li**    **Yaniv Plan**    **Özgür Yılmaz**
Department of Mathematics
The University of British Columbia
{b.joshi, xli, yaniv, oyilmaz}@math.ubc.ca

## Abstract

We consider the problem of recovering an unknown latent code vector under a known generative model. For a $d$-layer deep generative network $\mathcal{G} : \mathbb{R}^{n_0} \to \mathbb{R}^{n_d}$ with ReLU activation functions, let the observation be $\mathcal{G}(x) + \epsilon$ where $\epsilon$ is noise. We introduce a simple novel algorithm, Partially Linearized Update for Generative Inversion (PLUGIn), to estimate $x$ (and thus $\mathcal{G}(x)$). We prove that, when weights are Gaussian and layer widths $n_i \gtrsim 5^i n_0$ (up to log factors), the algorithm converges geometrically to a neighbourhood of $x$ with high probability. Note the inequality on layer widths allows $n_i > n_{i+1}$ when $i \geq 1$. To our knowledge, this is the first such result for networks with some contractive layers. After a sufficient number of iterations, the estimation errors for both $x$ and $\mathcal{G}(x)$ are at most in the order of $\sqrt{4^d n_0/n_d}\|\epsilon\|$. Thus, the algorithm can denoise when the expansion ratio $n_d/n_0$ is large. Numerical experiments on synthetic data and real data are provided to validate our theoretical results and to illustrate that the algorithm can effectively remove artifacts in an image.

## 1 Introduction

We consider the inverse problem of recovering an unknown vector $x^* \in \mathbb{R}^{n_0}$ from a noisy observation $y \in \mathbb{R}^{n_d}$ of the form

$$y = \mathcal{G}(x^*) + \epsilon, \tag{1}$$

where $\epsilon \in \mathbb{R}^{n_d}$ is noise and $\mathcal{G} : \mathbb{R}^{n_0} \to \mathbb{R}^{n_d}$ is a known $d$-layer feed-forward neural network with ReLU activation functions (in fact, it is straightforward to generalize to positively homogeneous activation functions, which we will elaborate later). Precisely, $\mathcal{G}$ has the form

$$\mathcal{G}(x) = \sigma(A_d\sigma(A_{d-1}\ldots\sigma(A_1 x)\ldots)), \tag{2}$$

where $\sigma(\cdot) = \max(\cdot, 0)$ is the ReLU activation function and $A_i \in \mathbb{R}^{n_i \times n_{i-1}}$ is the weight matrix in the $i$-th layer.

The inverse problem (1) has applications in, for example, signal denoising and signal compression [1, 2, 3]. In the case of denoising the typical goal is to recover a clean signal $y^*$ from its noisy observation $y^* + \epsilon$ and, in the case of compression, the goal is to find an efficient low dimensional representation of $y^*$. Traditional approaches for these problems often use priors on signals, for example, a sparsity prior with respect to a fixed basis or dictionary [4, 5, 6]. An emerging viewpoint is to use a generative prior that assumes the unknown signal $y^*$ is in the range of a deep generative model $\mathcal{G}$, i.e., $y^* = \mathcal{G}(x^*)$, and develop algorithms that can estimate the latent code vector $x^*$ from $y^* + \epsilon$, thus recovering $y^*$.

Recent advancements in training deep neural networks have shown that priors in the form of generative models can effectively map low dimensional vectors to the space of natural image classes

35th Conference on Neural Information Processing Systems (NeurIPS 2021).

[7, 8, 9]. Learned generative models can then be used as priors to solve various inverse problems including denoising [1, 10], compressive sensing [11, 12, 13, 14, 15, 16], phase retrieval [17], blind deconvolution [18, 19], low-rank matrix recovery [20] and have been shown to perform on par or outperform classical sparsity based approaches for these inverse problems. For example, in [1] the authors empirically showed that an end-to-end approach for denoising using a neural network that maps noisy patches in an image to noise-free ones achieves state-of-the-art performance and is on par with BM3D. Similarly, in [11] the authors empirically showed that for compressive sensing using generative prior, optimization of the empirical risk objective over the latent code space (of the generative prior) can recover a vector that effectively estimates the uncompressed signal with 5-10 times less measurements compared to Lasso in some cases.

Given that $y$ equals $\mathcal{G}(x^*)$, with possibly some additive noise, a standard way to estimate $x^*$, i.e., to invert $\mathcal{G}(x^*)$, would be to look for a minimizer of the program

$$\min_{x \in \mathbb{R}^{n_0}} \|y - \mathcal{G}(x)\|^2. \tag{3}$$

Unfortunately, this program is non-convex and to our knowledge there is no known efficient method that can achieve its global minimum in general. On the other hand, for generative networks with random (Gaussian) weight matrices, a line of papers showed that gradient-based algorithms can provably avoid local minima with high probability [10, 14, 17, 20]. In particular, [10] considers a random Gaussian noise model, and random Gaussian weight matrices $A_i$ which are highly expansive at each layer. Under these conditions, the authors show that the latent code vector $x^*$ can be accurately estimated using a gradient-based method that uses the (sub-)gradient updates given by

$$x^{k+1} = x^k - \eta (D_1 A_1)^\intercal (D_2 A_2)^\intercal \cdots (D_d A_d)^\intercal \left( \mathcal{G}(x^k) - y \right), \tag{4}$$

where $x^k$ is the $k$-th estimate, $\eta \in \mathbb{R}$ is step size, and $D_j$ is a diagonal matrix with entries that are either zero or one. Each $D_j$ zeros out the inactive rows of $A_j$ with respect to the estimate $x^k$ and so it is a function of $x^k$ (and $A_p$ for $p < j$). Thus, at each iteration all $D_j$ need to be updated.

In this paper, we show that we can drop the $D_j$ and still recover an accurate estimate of $x^*$. We propose the following iterative algorithm, Partially Linearized Update for Generative Inversion (PLUGIn), to estimate $x^*$:

$$\textbf{PLUGIn:} \qquad x^{k+1} = x^k - \eta A_1^\intercal A_2^\intercal \cdots A_d^\intercal \left( \mathcal{G}(x^k) - y \right). \tag{5}$$

This algorithm was inspired by previous work showing that latent vectors for non-linear single-index function can be approximately estimated by treating the function as linear [21, 22]. Similar to [10, 11, 12, 14, 17, 19, 23], for theoretical analysis, we assume the weight matrices are Gaussian. To show that the algorithm works more broadly, we conduct real data simulations. Applying the ideas in [21, 22] one can show that for any fixed $x^0$, the first iteration of (5) provides an unbiased estimate of $x^*$ with $\eta = 2^d$, which is generally not the case for the gradient descent estimates given by (4), see Section 3 for further details. Additionally, each iteration of PLUGIn maps the difference $\mathcal{G}(x^k) - y$ to the low dimensional latent code space using a *static* matrix $A_1^\intercal A_2^\intercal \dots A_d^\intercal$, which can be pre-multiplied and reused in subsequent iterations.

In addition to the randomness assumption on the weight matrices, when the width of the each layer satisfies $n_i \gtrsim 5^i n_0$ (up to log factors), noise $\epsilon$ is independent of weight matrices, and the step size $\eta$ is fixed, we show that the estimates provided by PLUGIn converge to a neighbourhood of $x^*$ geometrically with high probability. This result allows $n_i > n_{i+1}$ for $i \geq 1$ and thus can provide theoretical guarantees for estimating $x^*$ from (1) even when $\mathcal{G}$ has some contractive layers. To our knowledge, this is the first such result. Additionally, we show that the $k$-th recovery and reconstruction errors satisfy

$$\|x^k - x^*\| \lesssim \sqrt{\frac{4^d n_0}{n_d}} \|\epsilon\| \quad \text{and} \quad \|\mathcal{G}(x^k) - \mathcal{G}(x^*)\| \lesssim \sqrt{\frac{4^d n_0}{n_d}} \|\epsilon\|$$

with high probability when $k$ is large. Thus, PLUGIn can effectively denoise noisy observations satisfying (1) provided that the expansion ratio $n_d/n_0$ is large. These contributions are highlighted below:

1. We introduce a simple novel algorithm (5), called Partially Linearized Update for Generative Inversion (PLUGIn), for inverting a generative model. Note that each update of PLUGIn

can be implemented with two blackboxes. The first blackbox provides the $\mathcal{G}(x^k)$ for each iterate and the second blackbox provides the product of the weight matrices $A_1^\intercal A_2^\intercal \ldots A_d^\intercal$. Thus, given these two blackboxes the individual matrices are not needed.

2. Under relatively mild conditions, we show that for generative priors with possibly contractive layers, PLUGIn can effectively estimate the unknown latent code $x^*$ and the unknown signal $\mathcal{G}(x^*)$ from observation $y$ satisfying (1). The convergence is geometric and holds for a range of step sizes.

3. In the presence of arbitrary noise $\epsilon$, we show that the recovery and reconstruction error corresponding to the estimate $x^k$, for sufficiently large $k$, is approximately $\sqrt{4^d n_0/n_d}\|\epsilon\|$. In contrast to typical random noise assumptions, we only require the noise to be independent of the weight matrices. This allows stochastic noise (independent of weight matrices) as a special case if one can bound the norm of the stochastic noise with high probability.

**Organization of the paper**: In Section 1.1, we introduce notations used through the paper. In Section 2, we state the main results. In Section 3, we provide an outline of the proof. In Section 4, we provide numerics on synthetic and real data. In Section 5, we briefly discuss a generalization of activation functions, as well as limitations in our results.

## 1.1 Notations

For a positive integer $n$, let $[n] = \{1, 2, \ldots, n\}$. For a vector $x$, let $\|x\|$ be its Euclidean norm; for a matrix $A$, let $\|A\|$ be its operator norm; for a matrix $A$ and a set $\mathcal{T}$, let $\|A\|_\mathcal{T} := \sup_{x \in \mathcal{T} \setminus \{0\}} \frac{\|Ax\|}{\|x\|}$. Let $\mathcal{N}(0, t^2)$ be the normal distribution with mean zero and standard deviation $t$. Let $\mathbb{B}(x, r)$ be the Euclidean ball of radius $r$ centered at $x$ and let $\mathbb{B}^n(0, r)$ be the Euclidean ball in $\mathbb{R}^n$ with radius $r$, centered at origin. We use $C$ and $c$ to denote absolute constants (often $c$ for small ones and $C$ for large ones) which may vary from line to line. We also use $C_0, C_1$, etc., to denote particular absolute constants, which do not change throughout the paper.

## 2 Main Results

Let the generative model $\mathcal{G} : \mathbb{R}^{n_0} \to \mathbb{R}^{n_d}$ be defined as in Equation (2). We consider the inverse problem:

$$\text{Let: } x^* \in \mathbb{R}^{n_0}, \epsilon \in \mathbb{R}^{n_d}, y = \mathcal{G}(x^*) + \epsilon,$$
$$\text{Given: generative model } \mathcal{G} \text{ and observation } y,$$
$$\text{Estimate: latent code vector } x^* \text{ and } \mathcal{G}(x^*).$$

We assume:

A1. All activation functions are ReLU.

A2. Each $A_i \in \mathbb{R}^{n_i \times n_{i-1}}$ has i.i.d. $\mathcal{N}(0, 1/n_i)$ entries and $\{A_i\}_{i \in [d]}$ are independent.

A3. Layer widths (number of nodes in each layer) satisfy

$$n_i \geq C_0 5^i n_0 \log \left( \prod_{j=0}^{i-1} \frac{e n_j}{n_0} \right), \quad i \in [d] \tag{6}$$

for some (sufficiently large) absolute constant $C_0$.

A4. The noise $\epsilon$ does not depend on $\{A_i\}_{i \in [d]}$. (The noise may be deterministic or random.)

Let $x^0 \in \mathbb{R}^{n_0}$ and let $x^k$ be the result of applying $k$ iterations of PLUGIn (5). Under assumptions A1-A4, $x^k$ geometrically converges to a neighbourhood of $x^*$ (and also $\mathcal{G}(x^k)$ to a neighbourhood of $\mathcal{G}(x^*)$) for a range of step sizes near $2^d$. Precisely, we have the following theorem.

**Theorem 1.** *Let $\theta \in (0, \frac{4}{3})$ and let $\alpha = |1 - \theta| + \frac{1}{2}\theta \in (0, 1)$. Let $R$ be a positive number such that $\|x^0 - x^*\| \leq R$. Under assumptions A1-A4, the $k$-th estimate $x^k$ given by PLUGIn algorithm (5) with constant step size $\eta = \theta 2^d$ satisfies*

$$\|x^k - x^*\| \leq \alpha^k R + \frac{15\theta}{1 - \alpha} 2^d \sqrt{n_0/n_d}\|\epsilon\|, \text{ and}$$

$$\|\mathcal{G}(x^k) - \mathcal{G}(x^*)\| \le 3\alpha^k R + \frac{45\theta}{1-\alpha} 2^d \sqrt{n_0/n_d} \|\epsilon\|$$

*with probability at least $1 - 2(k+3)e^{-10n_0}$.*

When $\theta = 1$, Theorem 1 reduces to the following corollary.

**Corollary 1.** *Let $R$ be a positive number such that $\|x^0 - x^*\| \le R$. Under assumptions A1-A4, the $k$-th estimate $x^k$ given by PLUGIn algorithm (5) with constant step size $\eta = 2^d$ satisfies*

$$\|x^k - x^*\| \le 2^{-k} R + 30 \cdot 2^d \sqrt{n_0/n_d} \|\epsilon\|, \text{ and}$$
$$\|\mathcal{G}(x^k) - \mathcal{G}(x^*)\| \le 2^{-k}(3R) + 90 \cdot 2^d \sqrt{n_0/n_d} \|\epsilon\|$$

*with probability at least $1 - 2(k+3)e^{-10n_0}$.*

**Remark 1** (Allowed number of iterations and comparison of error bound summands). The probability in Theorem 1 (and also Corollary 1) decreases as the number of iterations increases. However, this is generally not a problem because we can take up to exponentially many iterations before this probability becomes trivial. In fact, if we require a probability of at least 99% on the error bounds, then we can take any $k \le k^\star(n_0) := \frac{1}{200} e^{10n_0} - 3$. Note that $k^\star$ grows exponentially in $n_0$ and it is already large when $n_0$ is relatively small[1].

The noise term in the error bounds (proportional to $\|\epsilon\|$) will dominate the other term (long) before $k^\star$ many iterations are reached, unless the noise is virtually zero. Indeed, in Corollary 1, if $2^{-k^\star} R \ge \sqrt{4^d n_0/n_d} \|\epsilon\|$, then the noise must satisfy

$$\|\epsilon\| < R\sqrt{\frac{n_d}{4^d n_0}} \left(\frac{1}{2}\right)^{k^\star} = 8R\sqrt{\frac{n_d}{4^d n_0}} \left(\frac{1}{2}\right)^{\exp(10n_0)/200}.$$

Note that $\left(\frac{1}{2}\right)^{\exp(10n_0)/200}$ is a double exponential and extremely small[2], so we will consider the noise as being virtually zero in this case. On the other hand, if $2^{-k^\star} R < \sqrt{4^d n_0/n_d} \|\epsilon\|$, then after at most $k^\star$ iterations, with probability at least 99%, the estimation errors for $x^*$ and $\mathcal{G}(x^*)$ will be at most $C\sqrt{4^d n_0/n_d} \|\epsilon\|$. The same argument can be made for Theorem 1 when $\alpha$ is close to $\frac{1}{2}$ (i.e., $\theta$ is close to 1) so that $\alpha^k$ decays fast and $\frac{1}{1-\alpha}$ is not close to being singular.

**Remark 2** (Contractive layers). In A3, (6) states a lower bound on $n_i$ with respect to the latent code dimension $n_0$ (up to log factors). While this bound strictly increases with layer depth $i$, it is not necessary for $n_i$ to always increase with $i$ (except in the first layer). For example, consider $n_i = \beta C_0 5^d n_0 d(2d - i)$ where $\beta$ is any fixed number such that $\beta C_0 \in \mathbb{N}$ and $\beta \ge 4 + \log C_0$. It is easy to see $n_1 > n_2 > \cdots > n_d$, and we can also verify (see Appendix D) that such $n_i$ satisfy (6). In this case, the network is contractive in each layer after the first, and Theorem 1 still applies.

**Remark 3** (Initialization may depend on random weight matrices). The results of the theorem can still hold when $x^0$ is chosen randomly, dependent on the weight matrices $A_i$. In this case, suppose that $\|x^0 - x^*\| \le R$ with probability at least $1 - \delta$. Then, the error bounds hold with probability at least $1 - 2(k+3)e^{-10n_0} - \delta$. This does not follow directly from the theorem as stated (which fixes $x^0$, then takes random weight matrices), but follows from the proof.

**Remark 4** (Comparison to guarantees for gradient-based method). Here we compare our results to the ones in [10], which uses (4) for iterations and considers a random noise model.

When the noise $\epsilon \sim \mathcal{N}(0, \frac{1}{n_d}\gamma^2 I_{n_d})$, the error bounds on the $k$-th PLUGIn iterate in Corollary 1, with high probability, reduces to

$$\max\{\|x^k - x^*\|, \|\mathcal{G}(x^k) - \mathcal{G}(x^*)\|\} \le C\left(2^{-k} R + 2^d \sqrt{n_0/n_d}\gamma\right).$$

Thus, we get geometric convergence with a rate of $1/2$ and after sufficiently many iterations, the errors are at most $C2^d \sqrt{n_0/n_d}\gamma$.

A similar result was shown in [10], which studies a gradient-based method for solving the inverse problem (1). They show that when the weight matrices are random Gaussian and sufficiently

---

[1] For example, $k^\star(5) \approx 2.59 \times 10^{19}$ and $k^\star(10) \approx 1.34 \times 10^{41}$.
[2] For example, $2^{-\exp(10)/200} \approx 7 \times 10^{-34}$.

expansive at each layer, the iterates of the gradient-based method converge to a neighborhood of the target signal $x^*$. The result holds for a fixed step size, dependent on $d$. After sufficiently many iterations, the iterates converge geometrically to a neighborhood of $x^*$ of radius at most on the order of $\sqrt{n_0/n_d}\gamma$, up to log factors. This rate of convergence takes the form $(1 - C/d^2)$, thus giving slower convergence for deeper nets. On the other hand, we note that dependence on $d$ is of relatively minor concern. Generative models usually have small depth in practice, our MNIST experiments (below) work well with depth 3, and typical applications use depth less than 8.

## 3  Proof Outline

Here we give a sketch for the proof of Theorem 1. For simplicity, we will only focus on analyzing one iteration of PLUGIn with step size $\eta = 2^d$. The complete proof can be found in appendices.

**The Special Case**

Let us first look at the special case where $d = 1$ and $\epsilon = 0$. The analysis here highlights some of the key ideas in our proofs, while its result Lemma 1 serves as a building block for proof in the general case. In this special case, PLUGIn with $\eta = 2^d$ reduces to

$$x^{k+1} = x^k - 2A^\intercal \left[ \sigma(Ax^k) - \sigma(Ax^*) \right]$$

where $\sigma = \mathrm{ReLU}$ and $A \in \mathbb{R}^{m \times n}$ is random with i.i.d. $\mathcal{N}\left(0, \frac{1}{m}\right)$ entries.

In fact, the first iterate provides an unbiased estimate of $x^*$ when $x^0$ does not depend on $A$. Indeed, the rotation invariance property of the Gaussian distribution may be leveraged to show [21, 22], for any fixed $x$,

$$\mathbb{E}A^\intercal \sigma(Ax) = \tfrac{1}{2}x. \tag{7}$$

For completeness, we also include a proof for (7) in Appendix A, Lemma 2. Applying (7) to the first iteration gives

$$\mathbb{E}x^1 = x^0 - 2\mathbb{E}A^\intercal \sigma(Ax^0) + 2\mathbb{E}A^\intercal \sigma(Ax^*)$$
$$= x^0 - x^0 + x^* = x^*$$

Thus, even the first iterate can be shown to be a good estimate by showing that $x^1$ concentrates around its mean. Further iterates are generally no longer unbiased estimators because they pick up complex dependence on the random matrix $A$. We overcome this by developing a series of uniform deviation inequalities, as below.

Let us suppose we have shown that, with high probability, $\|x^k - x^*\| \leq r$ for some (small) constant $r > 0$. Then we wish to show that $\|x^{k+1} - x^*\| \leq r/2$ with high probability. Notice that

$$-(x^{k+1} - x^*) = 2A^\intercal \left[ \sigma(Ax^k) - \sigma(Ax^*) \right] - (x^k - x^*)$$
$$\|x^{k+1} - x^*\| = \sup_{u \in \mathbb{S}^{n-1}} 2 \left\langle Au, \sigma(Ax^k) - \sigma(Ax^*) \right\rangle - \left\langle u, x^k - x^* \right\rangle$$
$$= 2 \sup_{u \in \mathbb{S}^{n-1}} Z(u, x^k; x^*)$$

where

$$Z(u, v; x^*) := \left\langle Au, \sigma(Av) - \sigma(Ax^*) \right\rangle - \tfrac{1}{2} \left\langle u, v - x^* \right\rangle.$$

We wish to bound the supremum of random process $Z(u, x^k; x^*)$ over $u \in \mathbb{S}^{n-1}$. However, this process is challenging to analyze since $x^k$ depends on $A$ when $k \geq 1$. To alleviate this dependency, we bound by the supremum of $Z(u, v; x^*)$ over $(u, v) \in \mathcal{T}^0 := \mathbb{B}^n(0, 1) \times \mathbb{B}(x^*, r)$ instead. It is worth noting that $Z(u, v; x^*)$ is centred, namely $\mathbb{E}Z(u, v; x^*) = 0$ for any fixed $(u, v)$. We now arrive at the estimate

$$\|x^{k+1} - x^*\| \leq 2 \sup_{(u,v) \in \mathcal{T}^0} Z(u, v; x^*) \quad \text{if } \|x^k - x^*\| \leq r. \tag{8}$$

The following Lemma 1 provides a bound on $\sup_{\mathcal{T}^0} Z(u, v; x^*)$. In fact, it is slightly more general because we replaced $\mathcal{T}^0$ with $\mathcal{T}_1 \times \mathcal{T}_2$ (this replacement is helpful when studying the general case $d > 1$). The proof of this lemma can be found in Appendix B. The proof idea is to first establish

that $Z(u, v; x^*)$ has mixed (sub-Gaussian and sub-exponential) tail increments through Bernstein's inequality, and then apply the result from [24], which provides a general bound for the supremum of random processes with mixed tail increments.

**Lemma 1.** *Let $\sigma = \mathrm{ReLU}$. Fix $w \in \mathbb{R}^n$ and let $A \in \mathbb{R}^{m \times n}$ have i.i.d. $\mathcal{N}\left(0, \frac{1}{m}\right)$ entries. Define*

$$Z(u, v; w) := \langle Au, \sigma(Av) - \sigma(Aw) \rangle - \tfrac{1}{2} \langle u, v - w \rangle.$$

*Suppose $\mathcal{T}_1, \mathcal{T}_2$ are sets (not depending on $A$) such that*

$$\mathcal{T}_1 = \mathcal{S}_1 \cap \mathbb{B}^n(0, \alpha) \quad \text{and} \quad \mathcal{T}_2 = \mathcal{S}_2 \cap \mathbb{B}(w, \alpha r)$$

*for some $q$-dimensional (affine) subspaces $\mathcal{S}_1, \mathcal{S}_2 \subseteq \mathbb{R}^n$ and real numbers $\alpha, r > 0$. Then for any $t \geq 1$,*

$$\sup_{\substack{u \in \mathcal{T}_1 \\ v \in \mathcal{T}_2}} |Z(u, v; w)| \leq C_1 \alpha^2 r \left( \sqrt{\frac{q}{m}} + \frac{q}{m} + \sqrt{\frac{t}{m}} + \frac{t}{m} \right)$$

*with probability at least $1 - e^{-t}$. Here $C_1 > 0$ is an absolute constant.*

We can apply Lemma 1 to estimate (8) (with $\mathcal{S}_1 = \mathcal{S}_2 = \mathbb{R}^n$) and get, for example,

$$\|x^{k+1} - x^*\| \leq 2C_1 r \left( \sqrt{\frac{n}{m}} + \frac{n}{m} + \sqrt{\frac{n}{m}} + \frac{n}{m} \right) \leq \tfrac{1}{2} r$$

with probability at least $1 - e^{-n}$, provided that $m \geq (16 C_1)^2 n$.

**The General Case**

Let us illustrate the proof idea with $d = 2$ (the extension to $d > 2$ is straightforward). Denote $x_1^k = \sigma(A_1 x^k)$ and $x_1^* = \sigma(A_1 x^*)$. By adding and subtracting $2 A_1^\mathsf{T}(x_1^k - x_1^*)$ we can write (5) with $\eta = 2^d$ as

$$\begin{aligned}
x^{k+1} - x^* &= x^k - x^* - 2^2 A_1^\mathsf{T} A_2^\mathsf{T} [\mathcal{G}(x^k) - \mathcal{G}(x^*) - \epsilon] \\
&= (x^k - x^*) - 2 A_1^\mathsf{T} \left( \sigma(A_1 x^k) - \sigma(A_1 x^*) \right) \\
&\quad + 2 A_1^\mathsf{T} \left[ (x_1^k - x_1^*) - 2 A_2^\mathsf{T} \left( \sigma(A_2 x_1^k) - \sigma(A_2 x_1^*) \right) \right] \\
&\quad + 2^2 A_1^\mathsf{T} A_2^\mathsf{T} \epsilon.
\end{aligned}$$

Similar to the special case above, we can get

$$\|x^{k+1} - x^*\| \leq \sup_{u \in \mathbb{S}^{n_0 - 1}} 2 Z_1(u, x^k) + \sup_{u \in \mathbb{S}^{n_0 - 1}} 2^2 Z_2(A_1 u, x_1^k) + 2^2 \|A_1^\mathsf{T} A_2^\mathsf{T} \epsilon\| \tag{9}$$

where (denote $x_0^* = x^*$)

$$Z_j(u, v) := \left\langle A_j u, \sigma(A_j v) - \sigma(A_j x_{j-1}^*) \right\rangle - \tfrac{1}{2} \left\langle u, v - x_{j-1}^* \right\rangle, \quad j = 1, 2.$$

Also assume that $\|x^k - x^*\| \leq r$, it remains to bound each term on the right hand side of (9). The first term can be bounded directly through Lemma 1 (with $t = 10 n_0$). The last term is also easy to bound by the randomness of $A_i$ (see Appendix C, Lemma 6), in which case we have $\|A_1^\mathsf{T} A_2^\mathsf{T} \epsilon\| \leq 15 \sqrt{n_0 / n_d} \|\epsilon\|$ with high probability.

Denote $\mathcal{G}_1(x) = \sigma(A_1 x)$ for $x \in \mathbb{R}^{n_0}$. For the second term, first notice that $\mathrm{range}(A_1)$ is a $n_0$-dimensional subspace in $\mathbb{R}^{n_1}$. Using the ideas from [11, 25], we can also show that $\mathrm{range}(\mathcal{G}_1)$ is contained in a union of $N$ many $n_0$-dimensional (affine) subspaces, where $N \leq (en_1 / n_0)^{n_0}$. Furthermore, let $\mathcal{E}$ be the event such that mappings $A_1, \mathcal{G}_1, \mathcal{G}$ all have Lipschitz constants being at most 3, then we can show (Appendix C, Lemma 8) that $\mathbb{P}(\mathcal{E}) \geq 1 - 3 e^{-10 n_0}$. Also on event $\mathcal{E}$ (note that $\|A_1\| \leq 3$ and $\|x_1^k - x_1^*\| \leq 3r$), we have

$$A_1 \mathbb{S}^{n_0 - 1} \subseteq \mathrm{range}(A_1) \cap \mathbb{B}^{n_1}(0, 3) = \mathcal{S}_1 \cap \mathbb{B}^{n_1}(0, 3) =: \mathcal{T}_1$$

$$x_1^k \in \mathrm{range}(\mathcal{G}_1) \cap \mathbb{B}(x_1^*, 3r) \subseteq \cup_{j \in [N]} \left( \mathcal{S}_{1,j} \cap \mathbb{B}(x_1^*, 3r) \right) =: \cup_{j \in [N]} \mathcal{T}_{2,j}$$

where $\mathcal{S}_1$ and $\mathcal{S}_{2,j}$ are $n_0$-dimensional (affine) subspaces. Applying Lemma 1 on each $\mathcal{T}_1 \times \mathcal{T}_{2,j}$, followed by a union bound over $j \in [N]$, we get (denote $\mathcal{T}_2 = \cup_{j \in [N]} \mathcal{T}_{2,j}$)

$$\sup_{\mathcal{T}_1 \times \mathcal{T}_2} Z_2(u, v) \leq C_1 (9r) \left( \sqrt{\frac{n_0}{n_2}} + \frac{n_0}{n_2} + \sqrt{\frac{t}{n_2}} + \frac{t}{n_2} \right)$$

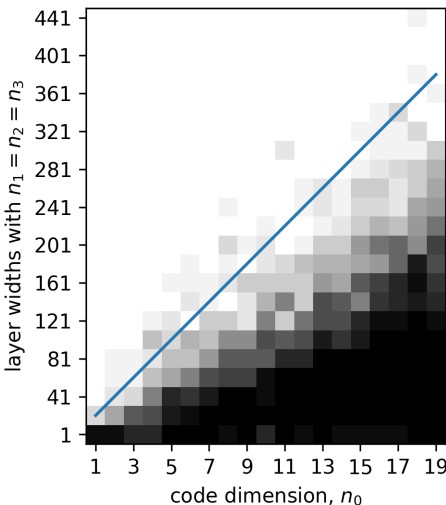

Figure 1: The empirical recovery probability from synthetic data in the noiseless case with code dimension level $n_0$ as a function of widths of the subsequent layers $n_1 = n_2 = n_3$. Each block correspond to the average from 20 independent trials. White blocks correspond to successful recovery and black blocks correspond to unsuccessful recovery. The area to the left of the line satisfies $n_3 > 20n_0$.

with probability (over $A_2$ and conditioning on $A_1$) at least $1 - Ne^{-t}$. By choosing $t = 2n_0 \log(en_1/n_0)$, we obtain a high probability bound for $\sup_{u \in \mathbb{S}^{n_0-1}} Z_2(A_1 u, x_1^k)$.

Finally, if $C_0$ is sufficiently large, we can thus show from (9) that, with high probability,

$$\|x^{k+1} - x^*\| \leq \frac{1}{2}\left(r + 30 \cdot 2^2 \sqrt{n_0/n_d}\|\epsilon\|\right).$$

## 4 Numerical Experiments

In this section, we provide numerical experiments on synthetic random data and MNIST images where the oberservations follow the model in (1). The first experiment illustrates a phase portrait that verifies Theorem 1 in the noiseless case. The second experiment illustrates that the PLUGIn algorithm (5) is stable to small dense noise and compares its performance to gradient descent. The third experiment illustrates decay rate of recovery error and reconstruction error of PLUGIn in the presence of noise and compares its performance to gradient descent. Lastly, the final experiments illustrates that PLUGIn can effectively remove dense unstructured random noise introduced to MNIST images. All experiments were conducted using Google Colaboratory.

In the synthetic experiments, we let the generative prior be a 3-layer neural network $\mathcal{G} : \mathbb{R}^{n_0} \to \mathbb{R}^{n_3}$ of the form $\mathcal{G}(z) = \sigma(A_3\sigma(A_2\sigma(A_1x)))$, where the entries of weight matrix $A_i \in \mathbb{R}^{n_i \times n_{i-1}}$ are sampled from $\mathcal{N}(0, 1/n_i)$. We sample the target latent code $x^*$ uniformly from $\mathbb{S}^{n_0-1}$, set the noise level as $\alpha \in \mathbb{R}$, and set the noise to be $\alpha\nu$ where $\nu$ is sampled uniformly from $\mathbb{S}^{n_3-1}$. Then we set $y = \mathcal{G}(x^*) + \alpha\nu$. We run PLUGIn and gradient descent each for 10,000 iterations, or until the difference in norm between consecutive iterates is less than $10^{-13}$ times the norm of the current iterate, whichever comes first, and set $\hat{x}$ to be the output. We use a fixed step size of $\eta = 3$ for PLUGIn; for gradient descent, we use a parameter of 1000 to move along the gradient descent direction computed using PyTorch [26].

For the first experiment, we fix $n_0 \in \{1, 2, \ldots, 19\}$, $n_1 \in \{1, 21, 41, \ldots, 441\}$, $n_3 = n_2 = n_1$, and $\alpha = 0$. We randomly initialize PLUGIn, setting $x^0$ to have independent standard normal entries. Note that the norm of $x^0$ concentrates around $\sqrt{n_0}$, and thus is generally far from the norm of the target latent vector. For each trial, we say PLUGIn successfully recovers the target latent code if

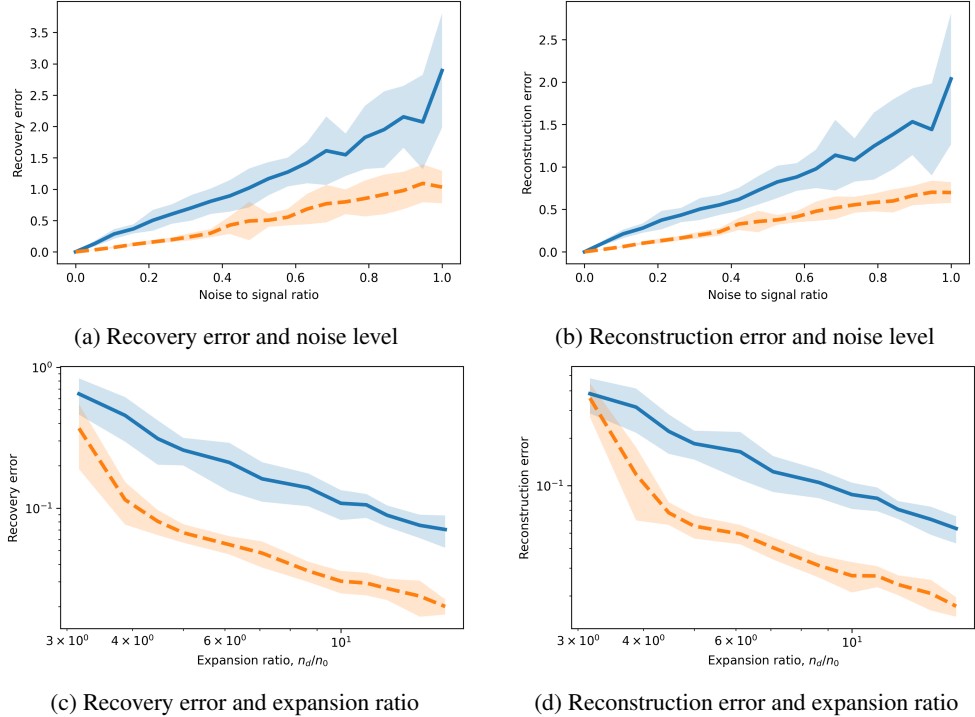

(a) Recovery error and noise level    (b) Reconstruction error and noise level

(c) Recovery error and expansion ratio   (d) Reconstruction error and expansion ratio

Figure 2: Comparison of PLUGIn (solid line) with gradient descent (dashed line). Panel (a) shows the dependence of relative recovery error with noise level-to-signal level from 20 independent trials. Panel (b) shows the dependence of relative reconstruction error and noise level-to-signal level. Panel (c) and (d) shows the dependence of relative recovery error and reconstruction error, respectively, with respect to the expansion ratio $n_3/n_0$.

$\|\hat{x} - x^*\| \leq 10^{-5}$. Figure 1 shows the fraction of successful recoveries from 20 independent trials using the from observation as described above. Black squares corresponds to no successful recovery and white squares correspond to 100% successful recovery.

For the second experiment, we fix $n_0 = 20$, $n_1 = 600$, $n_2 = 500$, $n_3 = 500$, and sample the noise level $\alpha$ uniformly in the interval $[0, 1]$. In figures 2a and 2b, the solid line corresponds to the performance of PLUGIn and the dotted line represents the performance of gradient descent. Figure 2a shows the empirical dependence of the the relative recovery error $\|\hat{x} - x^*\|/\|x^*\|$ on the noise-to-signal ratio given, given by $\alpha$, from 20 independent trials. Similarly, figure 2b shows the empirical dependence of the the relative reconstruction error $\|\mathcal{G}(\hat{x}) - \mathcal{G}(x^*)\|/\|\mathcal{G}(x^*)\|$ from 20 independent trials. The figures show that PLUGIn can be used to stably denoise signals using a generative prior with contractive layers.

For the third synthetic experiment, we fix $n_0 = 20$, noise level $\alpha = 0.1$ and sample $n_1 = n_2 = n_3$ in the interval $[200, 5000]$. Figures 2c and 2d show the empirical dependence of relative recovery error $\|\hat{x} - x^*\|/\|x^*\|$ and relative reconstruction error $\|\mathcal{G}(\hat{x}) - \mathcal{G}(x^*)\|/\|\mathcal{G}(x^*)\|$, respectively, on the expansion ratio $n_3/n_0$ from 20 independent trials. In figures 2c and 2d, the solid line corresponds to PLUGIn and the dotted line corresponds to gradient descent. The figures confirm the linear dependence in the recovery and reconstruction errors of PLUGIn, in log scale, with respect to $n_d/n_0$, provided $n_i$, for all $i \geq 1$, are sufficiently large compared to $n_0$.

We now empirically show that PLUGIn can effectively remove noise synthetically introduced to MNIST images and compare its performance to gradient descent. We trained a VAE [27] using Adam optimizer [28] with a learning rate of 0.001 and mini-batch size 100 on the MNIST dataset consisting of 60,000 number of $28 \times 28$ images of handwritten digits [29]. The latent code dimension of the trained network $\mathcal{G}$ is 20 and the decoder network in the VAE is a fully connected network with parameters $20 - 500 - 500 - 784$. Finally, a noisy image $y$ is generated via pixelwise addition of an image $y^*$ from the MNIST database and an noise vector $\nu \in \mathbb{R}^{784}$, i.e. $y = y^* + \nu$. In all MNIST

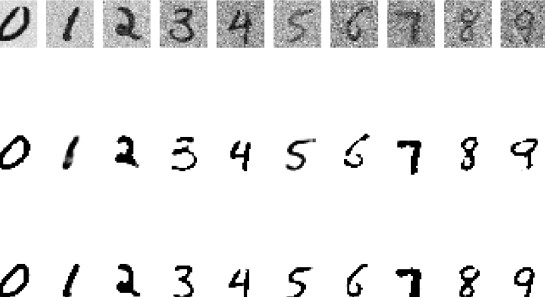

Figure 3: The figure shows the result of denosing an image using PLUGIn and gradient descent. The top row corresponds to noisy image. The second and third row corresponds to images recovered using PLUGIn and gradient descent, respectively.

experiments, we use a fixed step size of $\eta = 1/\gamma$ for PLUGIn, where $\gamma$ is the product of the operator norms of the weight matrices; for gradient descent, we use a parameter of 1000 to move along the gradient descent direction computed using PyTorch. Similar to the synthetic experiment, we run PLUGIn and gradient descent each for 10,000 iterations, or until the difference in norm between consecutive iterations is less than $10^{-13}$ times the norm of the current iterate, whichever comes first.

In figure 3, the images in the top row are the observations, the images in the second row and third row are the recovered images corresponding to PLUGIn and gradient descent, respectively. Figure 3 shows the result of using PLUGIn and gradient descent to remove noise $\nu$ sampled uniformly over the sphere $\mathbb{S}^{783}$ from observation $y$. The noise level $\alpha$ was adjusted so that the signal-to-noise ratio is in the interval $[50, 200]$, with 50 corresponding to the leftmost column and 200 corresponding to the rightmost one.

## 5    Final Remarks

**Extension to positively homogeneous activation functions**

Our proofs and results can be extended to generative model $\mathcal{G}$ with positively homogeneous activation functions[3] (such as Leaky ReLU). To see this, we modify the definition for $Z(u, v; w)$ in Lemma 1 to

$$Z(u, v; w) := \langle Au, \sigma(Av) - \sigma(Aw) \rangle - \lambda \langle u, v - w \rangle,$$

where $\lambda := \mathbb{E}\, g \cdot \sigma(g)$ with $g \sim \mathcal{N}(0, 1)$, and change the step size from $\eta = 2^d$ to $\eta = \lambda^{-d}$. The random process $Z(u, v; w)$ remains centred (Appendix A, Lemma 2) and one can prove a similar version of Lemma 1. Moreover, in the multi-layer case, the range at each layer is still contained in a union of $n_0$-dimensional (affine) subspaces in $\mathbb{R}^{n_i}$ (Appendix C, Lemma 7), so one can similarly prove a version of Theorem 1 in this case.

**Limitations**

In Theorem 1 we assumed that weight matrices are independent and Gaussian. Although this is a commonly used assumption in literature (e.g. [10, 11, 12, 14, 17, 19, 23]) for theoretical analysis, practitioners may wonder whether theory based upon this assumption is instructive towards predicting convergence when the algorithm is applied to generative models with learned weights. Our experiments with MNIST give some evidence that the algorithm does indeed generalize. Without the Gaussian assumption, one needs a heuristic for choosing the step size. We find that the inverse of the product of the norm of the weight matrices works well. Secondly, in numerical experiments, comparison between PLUGIn and gradient descent over synthetic data seems to suggest that PLUGIn has error with similar dependence on parameters to gradient descent, but with a larger constant.

---

[3]Activation functions $\sigma$ satisfies $\sigma(\alpha t) = \alpha\sigma(t)$ for all $\alpha \geq 0$ and $t \in \mathbb{R}$.

## Acknowledgments and Disclosure of Funding

Y. Plan is partially supported by an NSERC Discovery Grant (GR009284), an NSERC Discovery Accelerator Supplement (GR007657), and a Tier II Canada Research Chair in Data Science (GR009243). O. Yılmaz is partially supported by an NSERC Discovery Grant (22R82411) and Pacific Institute for the Mathematical Sciences (PIMS) CRG 33: High-Dimensional Data Analysis. B. Joshi is partially supported by the Pacific Institute for the Mathematical Sciences (PIMS). (The research and findings may not reflect those of the Institute.)

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
