# Appendices for *PLUGIn: A simple algorithm for inverting generative models with recovery guarantees*

## A  Some Results on Gaussian Matrices

Here we state some results on Gaussian Matrices, which will be used in the proofs later.

**Lemma 2** ([21, 22]). *Let $\sigma : \mathbb{R} \to \mathbb{R}$ be a positively homogeneous activation function. Let $A \in \mathbb{R}^{m \times n}$ have i.i.d. $\mathcal{N}\left(0, \frac{1}{m}\right)$ entries. Then for any $x \in \mathbb{R}^n$,*

$$\mathbb{E} A^{\mathsf{T}} \sigma(Ax) = \lambda x,$$

*where $\lambda := \mathbb{E}\, g \cdot \sigma(g)$ with $g \sim \mathcal{N}(0, 1)$. In particular, $\lambda = \frac{1}{2}$ when $\sigma$ is ReLU.*

*Proof.* Since $\sigma$ is positively homogeneous, we can assume (without loss of generality) $x \in \mathbb{S}^{n-1}$. Denote by $a_j^{\mathsf{T}}$ the $j$-th row of $A$. Then

$$\mathbb{E} A^{\mathsf{T}} \sigma(Ax) = \mathbb{E} \sum_{j=1}^{m} \sigma(a_j^{\mathsf{T}} x)\, a_j = m\, \mathbb{E} \sigma(a_1^{\mathsf{T}} x)\, a_1 = \mathbb{E} \sigma(a^{\mathsf{T}} x)\, a$$

where $a := \sqrt{m} a_1 \sim \mathcal{N}(0, I_n)$. Take an orthogonal matrix $U$ such that $Ux = \|x\| e_1 = e_1$ where $e_1 = (1, 0, \ldots, 0)^{\mathsf{T}}$. Note that by rotation invariance for standard Gaussian, $Ua$ and $a$ have the same distribution $\mathcal{N}(0, I_n)$, thus

$$\mathbb{E} \sigma(a^{\mathsf{T}} x)\, a = \mathbb{E} \sigma(a^{\mathsf{T}} U^{\mathsf{T}} e_1)\, U^{\mathsf{T}} U a = \mathbb{E} \sigma(a^{\mathsf{T}} e_1) U^{\mathsf{T}} a = U^{\mathsf{T}} \mathbb{E} \sigma(a^{\mathsf{T}} e_1) a = \lambda U^{\mathsf{T}} e_1 = \lambda x.$$

$\square$

The following theorem is the concentration of (Gaussian) measure inequality for Lipschitz functions. Here we only state a one-sided version, though it is more commonly stated with a two-sided one, i.e., $\mathbb{P}\left(|f(g) - \mathbb{E}f(g)| \geq t\right) \leq 2 \exp\left(-t^2/(2L_f^2)\right)$.

**Theorem 2.** *Let $f : \mathbb{R}^n \to \mathbb{R}$ be a Lipschitz function with Lipschitz constant $L_f$. Let $g \in \mathbb{R}^n$ be a random vector with independent $\mathcal{N}(0, 1)$ entries. Then, for all $t > 0$,*

$$\mathbb{P}\left(f(g) - \mathbb{E}f(g) \geq t\right) \leq \exp\left(-\frac{t^2}{2L_f^2}\right).$$

A proof of Theorem 2 can be found in [30, Chap. 8]. Based on this theorem, it is easy to prove the following results.

**Lemma 3.** *Let $A \in \mathbb{R}^{m \times n}$ have i.i.d. $\mathcal{N}(0, 1)$ entries.*

*(a) For any fixed point $s \in \mathbb{R}^n$, we have*

$$\mathbb{P}\left(\|As\| \geq \sqrt{m}\|s\| + \sqrt{t}\|s\|\right) \leq e^{-t/2}, \quad \forall t > 0.$$

*(b) For any fixed $k$-dimensional subspace $\mathcal{S} \subseteq \mathbb{R}^n$, we have*

$$\mathbb{P}\left(\|A\|_{\mathcal{S}} \geq \sqrt{m} + \sqrt{k} + \sqrt{t}\right) \leq e^{-t/2}, \quad \forall t > 0.$$

*Proof.* (a) Without loss of generality, assume $\|s\| = 1$. Then $As \sim \mathcal{N}(0, I_m)$ and by Jensen's inequality, $\mathbb{E}\|As\| \leq \sqrt{\mathbb{E}\|As\|^2} = \sqrt{m}$. The result follows immediately from Theorem 2 (with $f(g) = \|g\|$ and $g = As$).

(b) Let $U$ be an orthogonal matrix such that $U^{\mathsf{T}} \mathcal{S} = \operatorname{span}\{e_1, \ldots, e_k\} =: \mathcal{S}_0$, then $\|A\|_{\mathcal{S}} = \|AU\|_{\mathcal{S}_0}$. Also, since $AU$ has the same distribution as $A$ (by rotation invariance), we get

$$\mathbb{P}\left(\|A\|_{\mathcal{S}} \geq \sqrt{m} + \sqrt{k} + \sqrt{t}\right) = \mathbb{P}\left(\|A\|_{\mathcal{S}_0} \geq \sqrt{m} + \sqrt{k} + \sqrt{t}\right).$$

Notice that $\|A\|_{\mathcal{S}_0}$ is the operator norm for a particular sub-matrix (obtained by taking first $k$-columns) of $A$, so without loss of generality, we can assume $k = n$.

Let $f(A) = \|A\|$. Since $|f(A) - f(A')| \leq \|A - A'\|_F$, $f$ is 1-Lipschitz when viewed as a mapping from $\mathbb{R}^{mn}$ to $\mathbb{R}$. By Theorem 2,

$$\mathbb{P}\left( f(A) \geq \mathbb{E}f(A) + \sqrt{t} \right) \leq e^{-t/2}, \quad \forall t > 0.$$

The result follows since $\mathbb{E}\|A\| \leq \sqrt{m} + \sqrt{n}$ (see, e.g., [31, Section 7.3]). $\qquad\square$

## B  Preliminaries and Proof for Lemma 1

**Preliminaries**

For $\alpha \geq 1$, the $\psi_\alpha$-*norm* of a random variable $X$ is defined as

$$\|X\|_{\psi_\alpha} := \inf\{t > 0 : \mathbb{E}\exp(|X|^\alpha/t^\alpha) \leq 2\}.$$

We say $X$ is *sub-Gaussian* if $\|X\|_{\psi_2} < \infty$ and *sub-exponential* if $\|X\|_{\psi_1} < \infty$. The $\psi_2$ and $\psi_1$ norms are also called sub-Gaussian and sub-exponential norms respectively. Loosely speaking, a sub-Gaussian (or a sub-exponential) random variable has tail dominated by the tail of a Gaussian (or an exponential) random variable.

For independent, mean zero, sub-exponential random variables $X_1, \ldots, X_m$, their sum concentrates around zero. In particular, the following *Bernstein's Inequality* [31, Section 2.8] holds:

$$\mathbb{P}\left( \left| \sum_{i=1}^m X_i \right| \geq t \right) \leq 2\exp\left[ -c\min\left( \frac{t^2}{\sum_{i=1}^m \|X_i\|_{\psi_1}^2}, \frac{t}{\max_i \|X_i\|_{\psi_1}} \right) \right].$$

The above inequality also suggests that $\sum_{i=1}^m X_i$ has a mixed tail, i.e., a tail consisting of both a sub-Gaussian part and a sub-exponential part. In our proof, we will use the following result from generic chaining for mixed tail processes.

**Theorem 3** (Theorem 3.5 [24]). *If $(X_t)_{t \in T}$ has a mixed tail with respect to metric pair $(d_1, d_2)$, i.e.*

$$\mathbb{P}\left( |X_t - X_s| \geq \sqrt{u}d_2(t, s) + ud_1(t, s) \right) \leq 2e^{-u}, \quad \forall u \geq 0.$$

*Then there are constants $c, C > 0$ such that for any $u \geq 1$,*

$$\mathbb{P}\left( \sup_{t \in T} |X_t - X_{t_0}| \geq C(\gamma_2(T, d_2) + \gamma_1(T, d_1)) + c(\sqrt{u}\Delta_{d_2}(T) + u\Delta_{d_1}(T)) \right) \leq e^{-u}.$$

*Here $t_0$ is any fixed point in $T$, $\gamma_\alpha(T, d)$ is the $\gamma_\alpha$-functional and $\Delta_{d_i}$ is the diameter given by $\Delta_{d_i}(T) = \sup_{s, t \in T} d_i(s, t)$.*

The $\gamma_\alpha$-functional of $(T, d)$ is defined as

$$\gamma_\alpha(T, d) := \inf_{(T_n)} \sup_{t \in T} \sum_{n=0}^\infty 2^{n/\alpha} d(t, T_n), \tag{10}$$

where the infimum is taken with respect to all *admissible* sequences. A sequence $(T_n)_{n \geq 0}$ of subsets of $T$ is called *admissible* if $|T_0| = 1$ and $|T_n| \leq 2^{2^n}$ for all $n \geq 1$.

For our proof, we will use the following estimate on $\gamma_\alpha(T, d)$, which involves the generalized Dudley's integral [32, 24].

$$\gamma_\alpha(T, d) \leq C_{(\alpha)} \int_0^{\Delta_d(T)} (\log N(T, d, \varepsilon))^{1/\alpha} \, d\varepsilon, \tag{11}$$

where $C_{(\alpha)}$ is a constant depending only on $\alpha$ and $N(T, d, \varepsilon)$ is the *covering number*, i.e., the smallest number of balls (in metric $d$ and with radius $\varepsilon$) needed to cover set $T$.

**Proof for Lemma 1**

We recall the statement of Lemma 1 below.

**Lemma 1.** *Let $\sigma = \mathrm{ReLU}$. Fix $w \in \mathbb{R}^n$ and let $A \in \mathbb{R}^{m \times n}$ have i.i.d. $\mathcal{N}\left(0, \frac{1}{m}\right)$ entries. Define*

$$Z(u, v; w) := \langle Au, \sigma(Av) - \sigma(Aw) \rangle - \tfrac{1}{2} \langle u, v - w \rangle.$$

*Suppose $\mathcal{T}_1, \mathcal{T}_2$ are sets (not depending on $A$) such that*

$$\mathcal{T}_1 = \mathcal{S}_1 \cap \mathbb{B}^n(0, \alpha) \quad and \quad \mathcal{T}_2 = \mathcal{S}_2 \cap \mathbb{B}(w, \alpha r)$$

*for some $q$-dimensional (affine) subspaces $\mathcal{S}_1, \mathcal{S}_2 \subseteq \mathbb{R}^n$ and real numbers $\alpha, r > 0$. Then for any $t \geq 1$,*

$$\sup_{\substack{u \in \mathcal{T}_1 \\ v \in \mathcal{T}_2}} |Z(u, v; w)| \leq C_1 \alpha^2 r \left( \sqrt{\frac{q}{m}} + \frac{q}{m} + \sqrt{\frac{t}{m}} + \frac{t}{m} \right)$$

*with probability at least $1 - e^{-t}$. Here $C_1 > 0$ is an absolute constant.*

*Proof.* First, we establish that $Z(u, v; w)$ has a mixed tail.

Let $a_i^{\mathsf{T}}$ be the $i$-th row of $A$, then $a_i \sim \mathcal{N}(0, I_n/m)$. For $u \in \mathbb{B}^n(0, \alpha)$ and $v \in \mathbb{B}(w, \alpha r)$, define random variables

$$Z_{u,v}^i := \langle a_i, u \rangle \left[ \sigma(\langle a_i, v \rangle) - \sigma(\langle a_i, w \rangle) \right] - \tfrac{1}{2m} \langle u, v - w \rangle, \quad i \in [m].$$

We have $\mathbb{E} Z_{u,v}^i = 0$ by Lemma 2, and

$$Z_{u,v} := \sum_{i=1}^m Z_{u,v}^i = \langle Au, \sigma(Av) - \sigma(Aw) \rangle - \tfrac{1}{2} \langle u, v - w \rangle = Z(u, v; w).$$

For the increments of $Z_{u,v}^i$, we have

$$
\begin{aligned}
Z_{u,v}^i - Z_{u',v'}^i &= \langle a_i, u \rangle \, \sigma(a_i^{\mathsf{T}} v) - \tfrac{1}{2m} \langle u, v \rangle - \langle a_i, u' \rangle \, \sigma(a_i^{\mathsf{T}} v') + \tfrac{1}{2m} \langle u', v' \rangle \\
&\quad - \langle a_i, u - u' \rangle \, \sigma(a_i^{\mathsf{T}} w) + \tfrac{1}{2m} \langle u - u', w \rangle \\
&= \langle a_i, u \rangle \, \sigma(a_i^{\mathsf{T}} v) - \tfrac{1}{2m} \langle u, v \rangle - \left[ \langle a_i, u \rangle \, \sigma(a_i^{\mathsf{T}} v') - \tfrac{1}{2m} \langle u, v' \rangle \right] \\
&\quad + \left[ \langle a_i, u \rangle \, \sigma(a_i^{\mathsf{T}} v') - \tfrac{1}{2m} \langle u, v' \rangle \right] - \langle a_i, u' \rangle \, \sigma(a_i^{\mathsf{T}} v') + \tfrac{1}{2m} \langle u', v' \rangle \\
&\quad - \langle a_i, u - u' \rangle \, \sigma(a_i^{\mathsf{T}} w) + \tfrac{1}{2m} \langle u - u', w \rangle \\
&= \langle a_i, u \rangle \left[ \sigma(a_i^{\mathsf{T}} v) - \sigma(a_i^{\mathsf{T}} v') \right] - \tfrac{1}{2m} \langle u, v - v' \rangle \\
&\quad + \langle a_i, u - u' \rangle \left[ \sigma(a_i^{\mathsf{T}} v') - \sigma(a_i^{\mathsf{T}} w) \right] - \tfrac{1}{2m} \langle u - u', v' - w \rangle
\end{aligned}
$$

We can estimate its sub-exponential norm from Lemma 4, which gives

$$
\begin{aligned}
\| Z_{u,v}^i - Z_{u',v'}^i \|_{\psi_1} &\leq C_2 m^{-1} \left( \|u\| \|v - v'\| + \|u - u'\| \|v' - w\| \right) \\
&\leq C_2 \alpha m^{-1} \left( r \|u - u'\| + \|v - v'\| \right).
\end{aligned}
$$

By Bernstein's inequality,

$$\mathbb{P}\left( |Z_{u,v} - Z_{u',v'}| \geq t \right) \leq 2 \exp\left( -c \min\left( \frac{t^2}{d_2^2}, \frac{t}{d_1} \right) \right)$$

where the metrics $d_i$ are given by

$$d_2^2 = \frac{\alpha^2}{m} \left( r \|u - u'\| + \|v - v'\| \right)^2 \quad and \quad d_1 = \frac{\alpha}{m} \left( r \|u - u'\| + \|v - v'\| \right).$$

Therefore $(Z_{u,v})_{(u,v) \in \mathcal{T}}$ has a mixed tail with respect to the metric pair $(Cd_1, Cd_2)$ for some absolute constant $C$.

Next, we bound the supremum of $Z(u, v; w)$. Without loss of generality, we will assume that $q \geq 1$. (In fact, if $q = 0$, then $\mathcal{T}_1, \mathcal{T}_2$ are either empty set or singleton, in which case the result is trivial or follows directly from Bernstein's inequality).

Denote $\mathcal{T} := \mathcal{T}_1 \times \mathcal{T}_2$ and define a metric $d$ on $\mathcal{T}$ as

$$d\left((u,v),(u',v')\right) := r\|u - u'\| + \|v - v'\|.$$

It is easy to see that $d_2 = \frac{\alpha}{\sqrt{m}}d$ and $d_1 = \frac{\alpha}{m}d$. Also note that $\gamma_i(\mathcal{T}, td) = t\gamma_i(\mathcal{T}, d)$ from definition (10). We can assume that $\mathcal{S}_1$ is a subspace[4], then $Z_{0,v} = 0$ for $v \in \mathcal{T}_2$. Thus by Theorem 3, we have

$$\sup_{(u,v)\in\mathcal{T}} |Z_{u,v}| \lesssim \frac{\alpha}{\sqrt{m}}\gamma_2(\mathcal{T}, d) + \frac{\alpha}{m}\gamma_1(\mathcal{T}, d) + \sqrt{t}\frac{4\alpha^2 r}{\sqrt{m}} + t\frac{4\alpha^2 r}{m}$$

with probability at least $1 - e^{-t}$. It remains to estimate $\gamma_i(\mathcal{T}, d)$.
From (11) we have

$$\gamma_i(\mathcal{T}, d) \leq C_3 \int_0^{\Delta_d(\mathcal{T})} (\log N(\mathcal{T}, d, \varepsilon))^{1/i} \, d\varepsilon, \quad i = 1, 2.$$

Let $d_{\ell_2}$ be the Euclidean metric. Note that one can always obtain a $\varepsilon$-covering on $\mathcal{T}$ (with metric $d$) from the product set of a $\varepsilon/2$-covering on $\mathcal{T}_1$ (with metric $rd_{\ell_2}$) and a $\varepsilon/2$-covering on $\mathcal{T}_2$ (with metric $d_{\ell_2}$). Moreover, note that $\mathcal{T}_1$ is contained in a $q$-dimensional ball of radius $\alpha$ and $\mathcal{T}_2$ is contained in a $q$-dimensional ball of radius $\alpha r$. Hence

$$\begin{aligned}
N(\mathcal{T}, d, \varepsilon) &\leq N\left(\mathcal{T}_1, rd_{\ell_2}, \varepsilon/2\right) \cdot N\left(\mathcal{T}_2, d_{\ell_2}, \varepsilon/2\right) \\
&\leq N\left(\alpha\mathbb{B}^q, rd_{\ell_2}, \varepsilon/2\right) \cdot N\left(\alpha r\mathbb{B}^q, d_{\ell_2}, \varepsilon/2\right) \\
&= N\left(\mathbb{B}^q, d_{\ell_2}, \frac{\varepsilon}{2\alpha r}\right) \cdot N\left(\mathbb{B}^q, d_{\ell_2}, \frac{\varepsilon}{2\alpha r}\right) \\
&\leq \left(1 + \frac{4\alpha r}{\varepsilon}\right)^{2q}.
\end{aligned}$$

Here the last line uses estimate $N(\mathbb{B}^q, d_{\ell_2}, \varepsilon) \leq \left(1 + \frac{2}{\varepsilon}\right)^q$ for the covering number of unit balls (see e.g., [31, Section 4.2]).

Note the estimate[5] $\int_0^a \log\left(\frac{2a}{x}\right) dx = a(\log 2 + 1) < 2a$, we get

$$\gamma_1(\mathcal{T}, d) \leq C_3 \int_0^{4\alpha r} 2q \log\left(1 + \frac{4\alpha r}{\varepsilon}\right) d\varepsilon \leq 2C_3 q \int_0^{4\alpha r} \log\left(\frac{8\alpha r}{\varepsilon}\right) d\varepsilon \leq 16C_3\alpha rq.$$

Also note the inequality $\sqrt{\log(1+x)} < \sqrt{2}\log(1+x)$ for $x \geq 1$, we have

$$\begin{aligned}
\gamma_2(\mathcal{T}, d) &\leq C_3 \int_0^{4\alpha r} \sqrt{2q} \log^{\frac{1}{2}}\left(1 + \frac{4\alpha r}{\varepsilon}\right) d\varepsilon \\
&\leq 2C_3\sqrt{q} \int_0^{4\alpha r} \log\left(1 + \frac{4\alpha r}{\varepsilon}\right) d\varepsilon \\
&\leq 2C_3\sqrt{q} \int_0^{4\alpha r} \log\left(\frac{8\alpha r}{\varepsilon}\right) d\varepsilon \\
&\leq 16C_3\alpha r\sqrt{q}.
\end{aligned}$$

Therefore with probability at least $1 - e^{-t}$,

$$\sup_{(u,v)\in\mathcal{T}} |Z_{u,v}| \leq C_1\alpha^2 r \left(\sqrt{\frac{q}{m}} + \frac{q}{m} + \sqrt{\frac{t}{m}} + \frac{t}{m}\right).$$

$\square$

---

[4]If $\mathcal{S}_1$ is an affine subspace, let $q' = q + 1$ and let $\mathcal{S}_1'$ be the $q'$-dimensional subspace containing $\mathcal{S}_1$ (and origin). One can proceed with $\mathcal{S}_1'$ and $q'$ for the proof. Finally, notice that $\sqrt{\frac{q'}{m}} + \frac{q'}{m} \leq 2\left(\sqrt{\frac{q}{m}} + \frac{q}{m}\right)$, so this will give the same result with only a different absolute constant. (In fact, in our application of Lemma 1 for the multi-layer proof, $\mathcal{S}_1$ is chosen as $\text{range}(A_i \cdots A_1)$, which is always a subspace.)

[5]This comes from the indefinite integral $\int \log\left(\frac{a}{x}\right) dx = x \log\left(\frac{a}{x}\right) + x + C$.

**Lemma 4.** *Let $\sigma = \mathrm{ReLU}$. For $u, x, y \in \mathbb{R}^n$ and $g \sim \mathcal{N}(0, I_n)$, the (mean zero) random variable*

$$Z^g := \langle g, u \rangle \left[ \sigma(g^\mathsf{T} x) - \sigma(g^\mathsf{T} y) \right] - \tfrac{1}{2} \langle u, x - y \rangle$$

*has sub-exponential norm $\|Z^g\|_{\psi_1} \le C_2 \|u\| \|x - y\|$, where $C_2$ is an absolute constant.*

*Proof.* It is easy to see that $Z^g$ is mean zero from Lemma 2. Also from the following two properties of $\psi_1, \psi_2$-norms (see [31, Section 2.7]):

$$\|X - \mathbb{E}X\|_{\psi_1} \lesssim \|X\|_{\psi_1} \quad \text{and} \quad \|XY\|_{\psi_1} \le \|X\|_{\psi_2} \|Y\|_{\psi_2},$$

we have (note that $\sigma$ is 1-Lipschitz)

$$\|Z^g\|_{\psi_1} \lesssim \| \langle g, u \rangle \|_{\psi_2} \|\sigma(g^\mathsf{T} x) - \sigma(g^\mathsf{T} y)\|_{\psi_2} \lesssim \| \langle g, u \rangle \|_{\psi_2} \| \langle g, x - y \rangle \|_{\psi_2}.$$

The result follows by noting that $\| \langle g, u \rangle \|_{\psi_2} = \|g_1\|_{\psi_2} \|u\|$ where $g_1 \sim \mathcal{N}(0, 1)$. $\qquad\square$

## C   Proof for Theorem 1

**Additional notations:** We use $\mathbb{P}_{A_i}$ to denote that the probability is taken only with respect to $A_i$. In neural network $\mathcal{G} : \mathbb{R}^{n_0} \to \mathbb{R}^{n_d}$, let $\mathcal{G}_i : \mathbb{R}^{n_0} \to \mathbb{R}^{n_i}$ be the mapping that corresponds to the first $i$ layers, i.e. $\mathcal{G}_i(x) = \sigma(A_i \ldots \sigma(A_1 x) \ldots)$. For its weight matrices, let $\tilde{A}_0 = I_{n_0}$ and $\tilde{A}_i = A_i A_{i-1} \cdots A_1$ for $i \in [d]$.

*Proof of Theorem 1.* First we write

$$x^{k+1} - x^* = \theta \left( x^k - x^* - 2^d \tilde{A}_d^\mathsf{T} [\mathcal{G}(x^k) - y] \right) + (1 - \theta)(x^k - x^*).$$

For any fixed $r > 0$, using triangle inequality and Lemma 5 (with events $\mathcal{E}_i$ defined as in Lemma 5) we can conclude that if $\|x^k - x^*\| \le r$, then with probability at least $1 - \mathbb{P}(\mathcal{E}_1) - \mathbb{P}(\mathcal{E}_2) - 2e^{-10n_0}$,

$$\|x^{k+1} - x^*\| \le \frac{\theta}{2} \left( r + 30 \cdot 2^d \sqrt{\frac{n_0}{n_d}} \|\epsilon\| \right) + |1 - \theta| r = \alpha(r + \beta\varepsilon) \tag{12}$$

where

$$\alpha = \frac{\theta}{2} + |1 - \theta|, \quad \beta = \frac{\theta/2}{|1 - \theta| + \theta/2}, \quad \varepsilon = 30 \cdot 2^d \sqrt{n_0/n_d} \|\epsilon\|.$$

Now define a sequence $\{r_k\}_{k \in \mathbb{N}}$ such that $r_{k+1} = \alpha(r_k + \beta\varepsilon)$ and $r_0 = R$. We can find its general formula as follow:

$$r_{k+1} - \frac{\alpha\beta}{1 - \alpha}\varepsilon = \alpha \left( r_k - \frac{\alpha\beta}{1 - \alpha}\varepsilon \right) \quad \Rightarrow \quad r_k = \alpha^k \left( R - \frac{\alpha\beta}{1 - \alpha}\varepsilon \right) + \frac{\alpha\beta}{1 - \alpha}\varepsilon.$$

Next, by induction on $k$ (i.e., apply (12) with $r = r_k$ for $k = 0, 1, 2, \ldots$) we get

$$\|x^k - x^*\| \le r_k \le \alpha^k R + \frac{\alpha\beta}{1 - \alpha}\varepsilon, \quad k \in \mathbb{N}. \tag{13}$$

Notice that the events $\mathcal{E}_1, \mathcal{E}_2$ remain unchanged throughout iterations, so (13) holds with probability at least $1 - \mathbb{P}(\mathcal{E}_1) - \mathbb{P}(\mathcal{E}_2) - 2ke^{-10n_0}$.

Lastly, from Lemma 6 and Lemma 8 we know $\mathbb{P}(\mathcal{E}_i) \le 3e^{-10n_0}$ and $\|\mathcal{G}(x^k) - \mathcal{G}(x^*)\| \le 3\|x^k - x^*\|$ on $\mathcal{E}_2^c$. This completes the proof. $\qquad\square$

**Lemma 5.** *Fix $r > 0$ and assume assumptions A1-A4 hold. If $\|x^k - x^*\| \le r$, then after one iteration according to (5) with step size $\eta = 2^d$, we have*

$$\|x^{k+1} - x^*\| \le \frac{1}{2} \left( r + 30 \cdot 2^d \sqrt{\frac{n_0}{n_d}} \|\epsilon\| \right)$$

*with probability at least $1 - \mathbb{P}(\mathcal{E}_1) - \mathbb{P}(\mathcal{E}_2) - 2e^{-10n_0}$.*
*Here $\mathcal{E}_1, \mathcal{E}_2$ are the events*

$$\mathcal{E}_1 := \{\|\tilde{A}_d^\mathsf{T}\epsilon\| > 15\sqrt{n_0/n_d}\|\epsilon\|\} \quad \text{and} \quad \mathcal{E}_2 := \{\max(L_{\tilde{A}_i}, L_{\mathcal{G}_i}) > 3 \text{ for all } i \in [d]\}$$

*where $L_{\mathcal{G}_i}$ and $L_{\tilde{A}_i}$ denote the Lipschitz constants of $\mathcal{G}_i, \tilde{A}_i : \mathbb{R}^{n_0} \to \mathbb{R}^{n_i}$ respectively.*

*Proof.* For $x \in \mathbb{R}^{n_0}$, denote $x_0 = x$ and $x_i = \mathcal{G}_i(x)$ for $i \in [d]$. Then

$$
\begin{aligned}
x^{k+1} - x^* &= x^k - x^* - 2^d \tilde{A}_d^\mathsf{T}[\mathcal{G}(x^k) - \mathcal{G}(x^*) - \epsilon] \\
&= (x_0^k - x_0^*) - 2\tilde{A}_1^\mathsf{T}(x_1^k - x_1^*) \\
&\quad + 2\tilde{A}_1^\mathsf{T}\left[(x_1^k - x_1^*) - 2A_2^\mathsf{T}(x_2^k - x_2^*)\right] \\
&\quad + \dots \\
&\quad + 2^{d-1}\tilde{A}_{d-1}^\mathsf{T}\left[(x_{d-1}^k - x_{d-1}^*) - 2A_d^\mathsf{T}(x_d^k - x_d^*)\right] \\
&\quad + 2^d \tilde{A}_d^\mathsf{T}\epsilon
\end{aligned}
$$

thus we can write

$$
\begin{aligned}
\|x^{k+1} - x^*\| = \sup_{u \in \mathbb{S}^{n_0-1}} & \; 2\left(\langle A_1 u, x_1^k - x_1^* \rangle - \tfrac{1}{2}\langle u, x_0^k - x_0^* \rangle\right) \\
&+ 2^2\left(\langle A_2 \tilde{A}_1 u, x_2^k - x_2^* \rangle - \tfrac{1}{2}\langle \tilde{A}_1 u, x_1^k - x_1^* \rangle\right) \\
&+ \dots \\
&+ 2^d\left(\langle A_d \tilde{A}_{d-1} u, x_d^k - x_d^* \rangle - \tfrac{1}{2}\langle \tilde{A}_{d-1} u, x_{d-1}^k - x_{d-1}^* \rangle\right) \\
&- 2^d \langle u, \tilde{A}_d^\mathsf{T}\epsilon \rangle \\
\leq 2^d \|\tilde{A}_d^\mathsf{T}\epsilon\| &+ \sum_{i=0}^{d-1} 2^{i+1} \sup_{u \in \mathbb{S}^{n_0-1}} Z_{i+1}\left(\tilde{A}_i u, x_i^k\right)
\end{aligned}
$$

where

$$
Z_j(u,v) := \langle A_j u, \sigma(A_j v) - \sigma(A_j x_{j-1}^*) \rangle - \tfrac{1}{2}\langle u, v - x_{j-1}^* \rangle, \quad j \in [d].
$$

On event $\mathcal{E}_2^c$, $\forall i \in [d-1]$ we have

$$
\tilde{A}_i \mathbb{S}^{n_0-1} \subseteq \mathrm{range}(\tilde{A}_i) \cap \mathbb{B}^{n_i}(0,3) =: \mathcal{T}_1^i,
$$
$$
x_i^k \in \mathrm{range}(\mathcal{G}_i) \cap \mathbb{B}(x_i^*, 3r) =: \mathcal{T}_2^i.
$$

By Lemma 7, there are $N_{\mathcal{G}_i}$ many $n_0$-dimensional affine subspaces $\{\mathcal{S}_{i,j}\}$ such that

$$
\mathcal{T}_2^i \subseteq \cup_{j \in [N_{\mathcal{G}_i}]} \mathcal{T}_{2,j}^i \quad \text{where} \quad \mathcal{T}_{2,j}^i = \mathcal{S}_{i,j} \cap \mathbb{B}(x_i^*, 3r) \subseteq \mathbb{R}^{n_i} \text{ and } N_{\mathcal{G}_i} \leq \Phi_i := \prod_{j=1}^{i}\left(\frac{en_j}{n_0}\right)^{n_0}.
$$

For $i \in [d-1]$, apply Lemma 1 on $\mathcal{T}_1^i \times \mathcal{T}_{2,j}^i$ followed by a union bound over $j \in [N_{\mathcal{G}_i}]$, we get

$$
\sup_{\mathcal{T}_1^i \times \mathcal{T}_2^i} Z_{i+1}(u,v) \leq C_1(9r)\left(\sqrt{\frac{n_0}{n_{i+1}}} + \frac{n_0}{n_{i+1}} + \sqrt{\frac{t_{i+1}}{n_{i+1}}} + \frac{t_{i+1}}{n_{i+1}}\right)
$$

with probability (over $A_{i+1}$ and conditioning on $\{A_j\}_{j \in [i]}$) at least $1 - \Phi_i e^{-t_{i+1}}$.

Choose $t_{i+1} = 2\log \Phi_i = 2n_0 \sum_{j=1}^{i} \log\left(\frac{en_j}{n_0}\right)$, then we get

$$
\mathbb{P}_{A_{i+1}}\left(\sup_{\mathcal{T}_1^i \times \mathcal{T}_2^i} Z_{i+1}(u,v) \leq 9C_1 r \cdot 4\sqrt{\frac{2\log \Phi_i}{n_{i+1}}}\right) \geq 1 - e^{-\log \Phi_i}, \quad \forall i \in [d-1].
$$

Also for $i = 0$, applying Lemma 1 on $\mathbb{B}^{n_0}(0,1) \times \mathbb{B}(x^*, r)$, we get

$$
\sup_{\substack{u \in \mathbb{B}^{n_0}(0,1) \\ v \in \mathbb{B}(x^*, r)}} Z_1(u,v) \leq C_1 r \cdot 4\sqrt{\frac{10n_0}{n_1}}
$$

with probability (over $A_1$) at least $1 - e^{-10n_0}$.

Therefore under assumption A3 (with $C_0 \geq 160 \cdot 72^2 C_1^2$), we have

$$
\sum_{i=0}^{d-1} 2^{i+1} \sup_{u \in \mathbb{S}^{n_0-1}} Z_{i+1}\left(\tilde{A}_i u, x_i^k\right) \leq \frac{r}{72} + \sum_{i=1}^{d-1} 2^{i+1} \cdot \frac{r}{2}\sqrt{\frac{2}{160 \cdot 5^{i+1}}}
$$

$$= \frac{r}{72} + \frac{r}{2} \cdot \frac{1}{10} \sum_{i=1}^{d-1} \left( \frac{2}{\sqrt{5}} \right)^i$$

$$< \frac{r}{2} \cdot \frac{1}{10} \sum_{i=0}^{\infty} \left( \frac{2}{\sqrt{5}} \right)^i$$

$$< \frac{r}{2}$$

with probability at least $1 - \mathbb{P}(\mathcal{E}_2) - e^{-10n_0} - \sum_{i=1}^{d-1} e^{-\log \Phi_i}$.

The result follows by noting that (assume $C_0 \geq 160 \cdot 72^2$)

$$\log \Phi_i = n_0 \sum_{j=1}^{i} \log \left( \frac{e n_j}{n_0} \right) \geq n_0 i \log(e C_0) > 11 n_0 i,$$

so $\sum_{i \geq 1} e^{-\log \Phi_i} \leq \frac{e^{-11n_0}}{1 - e^{-11n_0}} < e^{-10n_0}$. Also note that on $\mathcal{E}_1^c$,

$$2^d \|\tilde{A}_d^\intercal \epsilon\| \leq 15 \cdot 2^d \sqrt{n_0/n_d} \|\epsilon\|.$$

$\square$

**Lemma 6.** *Under assumptions A2-A4, we have*

$$\mathbb{P} \left( \|A_1^\intercal A_2^\intercal \cdots A_d^\intercal \epsilon\| \geq 15 \sqrt{\frac{n_0}{n_d}} \|\epsilon\| \right) \leq 3 e^{-10n_0}.$$

*Proof.* Denote $s_i := A_{i+1}^\intercal \cdots A_d^\intercal \epsilon$ for $i \in [d-1]$ and $s_d := \epsilon$.

For $i \in [d]$, by Lemma 3(a) we have

$$\mathbb{P}_{A_i} \left( \sqrt{n_i} \|A_i^\intercal s_i\| \leq \sqrt{n_{i-1}} \|s_i\| + \sqrt{t_i} \|s_i\| \right) \geq 1 - e^{-t_i/2}, \quad \forall t_i > 0.$$

Choose $t_1 = 20 n_0$ and $t_j = n_{j-1}/4^{j-1}$ for $j > 1$, we get

$$\mathbb{P}_{A_1} \left( \|A_1^\intercal s_1\| \leq (1 + \sqrt{20}) \sqrt{\frac{n_0}{n_1}} \|s_1\| \right) \geq 1 - e^{-10n_0},$$

$$\mathbb{P}_{A_i} \left( \|A_i^\intercal s_i\| \leq (1 + 2^{-i+1}) \sqrt{\frac{n_{i-1}}{n_i}} \|s_i\| \right) \geq 1 - e^{-n_{i-1}/4^i}, \quad i > 1.$$

Thus with probability at least $1 - e^{-10n_0} - \sum_{i=2}^{d} e^{-n_{i-1}/4^i}$,

$$\|A_1^\intercal A_2^\intercal \cdots A_d^\intercal \epsilon\| \leq \left( 1 + \sqrt{20} \right) \sqrt{\frac{n_0}{n_1}} \cdot \prod_{i=2}^{d} \left( 1 + \frac{1}{2^{i-1}} \right) \sqrt{\frac{n_{i-1}}{n_i}}$$

$$\leq \left( 1 + \sqrt{20} \right) \sqrt{\frac{n_0}{n_d}} \cdot \prod_{i=1}^{\infty} \left( 1 + \frac{1}{2^i} \right)$$

$$< 15 \sqrt{n_0/n_d}$$

where the last inequality uses estimate[6] $\prod_{i=1}^{\infty} \left( 1 + \frac{1}{2^i} \right) \leq e$ and $(1 + \sqrt{20})e < 15$.

It remains to show $\sum_{i=2}^{d} e^{-n_{i-1}/4^i} \leq 2e^{-10n_0}$ for the desired probability bound. Note that by assumption A3 (assume $C_0 \geq 40$),

$$\frac{n_i}{4^{i+1}} \geq \frac{1}{4} C_0 n_0 \sum_{j=0}^{i-1} \log \left( \frac{e n_j}{n_0} \right) \geq 10 n_0 i.$$

Hence

$$\sum_{i=2}^{d} e^{-n_{i-1}/4^i} \leq \sum_{i=2}^{d} e^{-10n_0(i-1)} < \sum_{i=1}^{\infty} e^{-10n_0 i} = \frac{e^{-10n_0}}{1 - e^{-10n_0}} < 2e^{-10n_0}.$$

$\square$

---

[6]For $\alpha > 0$, estimate $\sum_{j=1}^{\infty} \log \left( 1 + \alpha 2^{-j} \right) \leq \sum_{j=1}^{\infty} \alpha 2^{-j} = \alpha$ holds, thus $\prod_{j=1}^{\infty} \left( 1 + \frac{\alpha}{2^j} \right) \leq e^\alpha$.

With ReLU (or positively homogeneous) activation functions, the range of neural network (in each layer) is contained in a union of affine subspaces. The following lemma, which is based on ideas and results in [11], gives a precise statement of this.

**Lemma 7.** *Assume A1 and $\min_{j\in[d]}\{n_j\} \geq n_0$, then for $i \in [d]$, $\mathrm{range}(\mathcal{G}_i)$ is contained in a union of affine subspaces. Precisely,*

$$\mathrm{range}(\mathcal{G}_i) \subseteq \cup_{j\in[N_{\mathcal{G}_i}]}\mathcal{S}_{i,j} \quad \text{where} \quad N_{\mathcal{G}_i} \leq \prod_{j=1}^{i}\left(\frac{en_j}{n_0}\right)^{n_0}.$$

*Here each $\mathcal{S}_{i,j}$ is some $n_0$-dimensional affine subspace (which depends on $\{A_l\}_{l\in[i]}$) in $\mathbb{R}^{n_i}$.*

*Proof.* The theory on hyperplane arrangements [25, Chapter 6.1] tells us that $n$ hyperplanes in $\mathbb{R}^k$ (assume $n \geq k$) partition the space $\mathbb{R}^k$ into at most $\sum_{j=0}^{k}\binom{n}{j}$ regions[7].

Also for $k \in [n]$,

$$\sum_{j=0}^{k}\binom{n}{j} \leq \sum_{j=0}^{k}\frac{n^j}{j!} \leq \sum_{j=0}^{k}\frac{k^j}{j!}\left(\frac{n}{k}\right)^j \leq \left(\frac{n}{k}\right)^k\sum_{j=0}^{\infty}\frac{k^j}{j!} = \left(\frac{en}{k}\right)^k.$$

So consider $\mathrm{range}(\mathcal{G}_1) = \{\sigma(A_1 x) : x \in \mathbb{R}^{n_0}\}$. Denote by $a_j^1$ ($j \in [n_1]$) the rows of $A_1$ and let $H$ be the set of hyperplanes $H := \cup_{j\in[n_1]}\{x : \langle a_j^1, x\rangle = 0\}$. Then $H$ partitions $\mathbb{R}^{n_0}$ into at most $(en_1/n_0)^{n_0}$ regions. Note that $\sigma$ is linear in each of these regions (thus the mapping $\mathcal{G}_1$ is linear in each region), so $\mathrm{range}(\mathcal{G}_1)$ is contained in at most $(en_1/n_0)^{n_0}$ many $n_0$-dimensional (affine) subspace.

The result then follows by induction. $\qquad\square$

The following lemma shows that the network $\mathcal{G}$ in our model is Lipschitz with high probability. This may be an interesting result on its own.

**Lemma 8.** *For mappings $\mathcal{G}_i$, $\tilde{A}_i : \mathbb{R}^{n_0} \to \mathbb{R}^{n_i}$, let $L_{\mathcal{G}_i}$ and $L_{\tilde{A}_i}$ be their Lipschitz constants respectively. Under assumptions A1-A3, we have*

$$\mathbb{P}\left(\max\{L_{\tilde{A}_i}, L_{\mathcal{G}_i}\} \leq 3 \text{ for all } i \in [d]\right) \geq 1 - 3e^{-10n_0}.$$

*Proof.* Denote $\tilde{\mathcal{R}}_0 = \mathcal{R}_0 = \mathbb{R}^{n_0}$ and

$$\mathcal{R}_j = \mathrm{range}(\mathcal{G}_j) - \mathrm{range}(\mathcal{G}_j), \quad \tilde{\mathcal{R}}_j = \mathcal{R}_j \cup \mathrm{range}(\tilde{A}_j), \quad j \in [d].$$

Note that $\tilde{A}_j$ is linear, so $\mathrm{range}(\tilde{A}_j)$ is a subspace in $\mathbb{R}^{n_i}$ with dimension at most $n_0$.

Since $\sigma$ is 1-Lipschitz, we have

$$\begin{aligned}
\|\mathcal{G}_i(x) - \mathcal{G}_i(x')\| &= \|\sigma(A_i\mathcal{G}_{i-1}(x)) - \sigma(A_i\mathcal{G}_{i-1}(x'))\| \\
&\leq \|A_i\left(\mathcal{G}_{i-1}(x) - \mathcal{G}_{i-1}(x')\right)\| \\
&\leq \|A_i\|_{\mathcal{R}_{i-1}}\|\mathcal{G}_{i-1}(x) - \mathcal{G}_{i-1}(x')\|.
\end{aligned}$$

Hence

$$\|\mathcal{G}_i(x) - \mathcal{G}_i(x')\| \leq \left(\prod_{l=1}^{i}\|A_l\|_{\tilde{\mathcal{R}}_{l-1}}\right)\|x - x'\|, \quad \forall i \in [d].$$

Similarly,

$$\|\tilde{A}_i x - \tilde{A}_i x'\| \leq \left(\prod_{l=1}^{i}\|A_l\|_{\tilde{\mathcal{R}}_{l-1}}\right)\|x - x'\|, \quad \forall i \in [d].$$

By Lemma 7, $\mathrm{range}(\mathcal{G}_i)$ is contained in a union of $N_{\mathcal{G}_i}$ many $n_0$-dimensional affine subspaces, so $\mathcal{R}_i$ is contained in a union of at most $N_{\mathcal{G}_i}^2$ many $2n_0$-dimensional affine subspaces. Since every

---

[7]Such regions are also called $k$-faces or $k$-cells. Relative to each of the $n$ hyperplanes, all points inside a region are on the same side.

$2n_0$-dimensional affine subspaces in $\mathbb{R}^{n_i}$ is also contained in a $(2n_0 + 1)$-dimensional subspace, we can further write this as

$$\tilde{\mathcal{R}}_i = \mathcal{R}_i \cup \text{range}(\tilde{A}_i) \subseteq \cup_{j \in [N_{\mathcal{G}_i}^2 + 1]} \mathcal{S}_{i,j} \quad \text{where} \quad N_{\mathcal{G}_i} \leq \Phi_i := \prod_{j=1}^{i} \left( \frac{en_j}{n_0} \right)^{n_0},$$

and each $\mathcal{S}_{i,j}$ is a $(2n_0 + 1)$-dimensional subspace in $\mathbb{R}^{n_i}$.

Thus by Lemma 3(b) and union bound we have, for $i \in [d-1]$,

$$\mathbb{P}_{A_{i+1}} \left( \sqrt{n_{i+1}} \|A_{i+1}\|_{\tilde{\mathcal{R}}_i} \geq \sqrt{n_{i+1}} + \sqrt{2n_0 + 1} + \sqrt{t_i} \right) \leq (\Phi_i^2 + 1) e^{-t_i/2}, \quad \forall t_i > 0.$$

Choose $t_i = 26 \log \Phi_i = 26 n_0 \sum_{j=1}^{i} \log(\frac{en_j}{n_0}) > 2n_0 + 1$ we get

$$\mathbb{P}_{A_{i+1}} \left( \|A_{i+1}\|_{\tilde{\mathcal{R}}_i} \geq 1 + 2\sqrt{\frac{26 \log \Phi_i}{n_{i+1}}} \right) \leq e^{-10 \log \Phi_i}.$$

Under assumption A3 (with $C_0 \geq 2^2 \cdot 26$), this implies

$$\mathbb{P}_{A_{i+1}} \left( \|A_{i+1}\|_{\tilde{\mathcal{R}}_i} \geq 1 + \frac{1}{2^{i+1}} \right) \leq e^{-10 \log \Phi_i}, \quad i \in [d-1].$$

Also by Lemma 3(b) with $t = 20 n_0$ and assumption A3 (assume $C_0 \geq 2^2 \cdot 26$), we have

$$\mathbb{P}_{A_1} \left( \|A_1\|_{\tilde{\mathcal{R}}_0} \geq 1 + \frac{1}{2} \right) \leq e^{-10 n_0}.$$

Therefore with probability at least $1 - e^{-10 n_0} - \sum_{i=1}^{d-1} e^{-10 \log \Phi_i}$,

$$\forall i \in [d], \quad \prod_{l=1}^{i} \|A_l\|_{\tilde{\mathcal{R}}_{l-1}} \leq \prod_{l=1}^{i} \left( 1 + \frac{1}{2^l} \right) \leq \prod_{l=1}^{\infty} \left( 1 + \frac{1}{2^l} \right) < 3.$$

Finally, note that $\log \Phi_i \geq i n_0$, so we have $\sum_{i=1}^{d-1} e^{-10 \log \Phi_i} \leq \sum_{i=1}^{\infty} e^{-10 n_0 i} < 2 e^{-10 n_0}$. This completes the proof. $\square$

## D  An Example of $n_i$

Here we show if $n_i = \beta C_0 5^d n_0 d(2d - i)$ where $\beta$ is any fixed number such that $\beta C_0 \in \mathbb{N}$ and $\beta \geq 4 + \log C_0$, then $n_i$ satisfy (6).

In fact, note that $2 \log d < d$ and $\log(2\beta) < \beta$, we have

$$
\begin{aligned}
\log \left( \prod_{j=0}^{i-1} \frac{en_j}{n_0} \right) &= 1 + \sum_{j=1}^{i-1} \log \left( \frac{en_j}{n_0} \right) \\
&\leq 1 + (d-1) \log \left( e\beta C_0 5^d \cdot 2d^2 \right) \\
&= 1 + (d-1)[d \log 5 + 2 \log d + \log(eC_0)] + (d-1) \log(2\beta) \\
&< 1 + d(d-1)[\log 5 + 1 + \log(eC_0)] + (d-1)\beta \\
&\leq \beta + d(d-1)\beta + (d-1)\beta \\
&= \beta d^2.
\end{aligned}
$$

Since $n_i \geq C_0 5^d n_0 (\beta d^2)$, it is easy to see that $n_i$ satisfy (6).

*Remark: A similar argument as above can also show that $n_i = \beta C_0 5^i n_0 i^2$ satisfy (6).*

## E  Code Link

Codes for numerical experiments are available at https://github.com/babhrujoshi/PLUGIn.