# OpenReview forum: "PLUGIn: A simple algorithm for inverting generative models with recovery guarantees"
_NeurIPS.cc/2021/Conference — NeurIPS 2021 Spotlight_

### Official Review · Reviewer_67vh · 2021-07-14

**Rating:** 8
**Confidence:** 3

**Summary:**

This paper studies the question of inverting a generative model; formally, given relu network G and a noisy observation y = G(x^*) + eps, approximately recover x. Similar to existing works in this literature, they assume the weight matrices in G are all Gaussian, suitably scaled. But whereas previous works needed to assume expansivity, namely that at each layer the ratio between output and input dimensions is at least some universal constant (or more), this paper only requires that the ratio between the output dimension of any layer L and the input dimension of *the network itself* scales at least exponentially in L. In particular, they can handle networks where some of the intermediate layers are contractive, as long as the network is expansive "on average" across layers.

In this setting, they analyze the update rule x' <- x - eta M^T(G(x) - y), where M is the product of the weight matrices, and show that it converges exponentially quickly to a neighborhood of x^*. Their assumption on eps is very mild: it just needs to be independent of the randomness of the weight matrices.

The motivation for this update rule is that when eps = 0, after a single step from an arbitrary initialization, the next iterate is equal to x^* in expectation with respect to the randomness of the weight matrices. This is not true for subsequent iterates as they depend on this randomness, but one can nevertheless look at the norm of the distance between any iterate and x^*. To break the dependence, the author(s) upper bound this by the sum of suprema of certain random processes corresponding to the different layers of the network. They give a high probability bound for these suprema via a chaining argument. The key fact that lets them avoid the need for expansivity is that these processes are over points that lie in a union of subspaces whose dimension is given by the *input dimension of the network*.

**Limitations And Societal Impact:**

There does not appear to be potential negative societal impact. Additionally, the authors clearly acknowledge the primary limitation of the result is the Gaussianity assumption, which is standard in this literature.

**Main Review:**

Inverting generative models is a very clean and well-motivated algorithmic challenge, and this paper contributes many interesting new ideas for this problem. From the point of view of the algorithm, existing approaches for this problem rely on showing that that the landscape for the square loss L(x) = ||G(x) - y||^2 is benign with high probability for expansive networks, so that gradient descent with respect to L converges to an approximate inverse for y. In this context, it is quite innovative to study the alternative update rule in this paper, and the fact that it works just as well for this problem and can even handle non-expansive networks is very interesting.

From the point of view of the analysis, existing works are based on verifying that a tall and skinny Gaussian weight matrix whp satisfies a certain "weight distribution condition" which one could also interpret as a bound on the supremum of some random process, this is by definition a condition that must hold *layer-by-layer* for the Hand-Voroninski-style analysis to apply. In contrast, the random processes that this paper considers are defined in terms of *sub-networks* of G rather than individual layers, which is why one can hope to get away with "expansivity on average" rather than strict expansivity.

Additionally, the technical details are well-written, and the example calculations in the first nine pages give the reader a good sense for how their proof works at a high level. I would enthusiastically recommend acceptance.

Minor comments:
- What are the obstacles for handling the setting where one only has compressive measurements of y?
- Caption for Figure 2 should be self-contained, i.e. should write that solid line is PLUGin and dotted line is gradient descent.
- In the display below line 462 of supplement, fix formatting for absolute values in |\sum^m_{i=1} X_i|
- Line 517 of supplement: capitalize "proof"
- Line 559 of supplement: "result" -> "results"

**Time Spent Reviewing:**

5

---

> ### Author Response · Authors · 2021-08-10
> **Response to Reviewer 67vh**
>
> We thank the reviewer for their review and comments. Please find our responses below:
>
> **Compressive measurements:** A result for the compressive measurement case can be readily obtained using similar techniques as the ones provided in the paper. In this case, we would consider the activation function of the last layer to be the identity function and the weight matrix of the last layer to be the compressive measurement matrix. In the random case, the recovery result would require a sample complexity on the order of $O(5^dn_0)$, up to log factors. Here, $n_0$ is the dimension of the latent code vector and $d$ is depth of the network. We believe, as an artifact of the proof technique, this sample complexity is optimal in $n_0$ but sub-optimal in $d$. We leave improving this dependence on $d$ to future work.
>
> **Typos and Caption:** We will fix them in the revised paper. Thank you again for pointing them out.

---

> > ### Comment · Reviewer_67vh · 2021-08-29
> > **Thanks for the response!**
> >
> > Thanks for clarifying regarding compressive measurements! I continue to believe that this is a very strong paper and enthusiastically recommend acceptance.

---

### Official Review · Reviewer_yoTL · 2021-07-14

**Rating:** 7
**Confidence:** 4

**Summary:**

This paper proposes an algorithm to solve the problem of inverting random generative models from noisy measurements where the weights of the model are drawn from $\mathcal{N}(0, 1/n)$ where n is the width of the corresponding layer, and the activation function is a ReLU unit.

Their proposed algorithm "Partially Linearized Update for Generative Inversion" (PLUGIn) is a simple iterative algorithm where each iterate is given by $x^{k+1} = x^k - \eta A_1^T \dots A_d^T (\mathcal{G}(x^k) - y)$. The authors demonstrate that

1. PLUGIn requires 2 blackboxes, (1) $\mathcal{G}(x^k)$ for every $x^k$ and (2) $ A_1^T \dots A_d^T $.
2. They show that for generative priors with possibly contractive layers, PLUGIn can effectively estimate the unknown signal from noisy observations, at a geometric rate.
3. In the presence of noise independent of the weight vector, the reconstruction error corresponding to $x^k$ for large enough $k$ goes as $\sqrt{4^d n_0/n_d}\| \epsilon \|$.

The proof of the result relies on simple properties of Gaussian matrices. In the case of the first layer it is enough to see that $E[A^T \sigma(Ax)] = 1/2 x$ and so with $\eta = 2$ we are done, for subsequent layers one needs to account for bias that creeps in and increase $\eta$ layer-wise. The authors overcome this by developing a series of uniform deviation inequalities, and bounding the resulting error.

**Limitations And Societal Impact:**

This is a theoretical paper and of limited societal impact

**Main Review:**

I think this is a good paper and makes progress in understanding when and how generative models are invertible. Prior work had intermediate layers with diagonal matrices, and could not tolerate noise that was only independent of the weight matrices.

I have read the author response.


**Time Spent Reviewing:**

2 hours

---

> ### Author Response · Authors · 2021-08-10
> **Response to Reviewer yoTL**
>
> We thank the reviewer for their review.

---

### Official Review · Reviewer_eL82 · 2021-07-16

**Rating:** 7
**Confidence:** 2

**Summary:**

This paper introduces an algorithm for recovering an unknown latent code vector under a specific generative model: a deep generative network with ReLU activation functions. Innovation of paper compared with [10] is to show that one drops the $D_j$ and still recover an accurate estimate of $x^*$. Theoretical results on geometrical convergence to a neighborhood of $x^{*}$ are confirmed by experiments on synthetic random data and MNIST images.

**Limitations And Societal Impact:**

Yes, the authors adequately addressed the limitations and potential negative societal impact of their work.

**Main Review:**

Unfortunately, I don't familiar quite well with generalized Dudley’s integral theory, so the questions are about numerical experiments:
`1. Did you try to check your method for other image datasets (for example CIFAR10)?
2. Figures 2c and 2d show that gradient descent numerically faster converges compared with PLUGIn. Does your method work much faster than SGD? Evaluations on time comparison for image/synthetic datasets and additional comments on the performance of PLUGIn compared with SGD are recommended.

**Time Spent Reviewing:**

4

---

> ### Author Response · Authors · 2021-08-10
> **Response to Reviewer eL82**
>
> Thank you for your review and suggestions. Please find our responses below:
>
> **Experiments on image dataset besides MNIST:** The paper is primarily focused on theoretical exposition of the PLUGIn algorithm. For numerical experiments, we provide preliminary positive results on its performance on MNIST dataset. We agree that further experiments on other datasets would be beneficial in demonstrating the applicability of the PLUGIN algorithm. We leave this to future work.
>
> **Comparison to SGD:** A natural choice of program to apply the stochastic gradient descent algorithm, in the context of the inverse problem considered in the paper, is
> $$ \\min_{x\\in\\mathbb{R}^{n_0}} \\|y - \\mathcal{G}(x)\\|_2^2,$$
> which is equation (3) in the paper. However, the objective in the above problem is not separable in entries of $x$ because the generator $\\mathcal{G}$ is a fully-connected neural network. As a result, we don't believe SGD can be efficiently implemented in this context and so a comparison to SGD is not provided.

---

### Decision · Program_Chairs · 2021-09-27

**Decision:**

Accept (Spotlight)

**Comment:**

This paper studies the problem of inverting a generative model. More precisely, we are given a ReLU network $G$ and we observe $y = G(x^*) + \epsilon$ and the goal is to recovery $x^*$. The paper gives strong provable guarantees under much milder assumptions than in earlier work. Earlier work of Hand and Voroninski gives recovery guarantees when the weights matrices in the network are Gaussian and each layer is expansive. This paper is able to weaken the former condition to an expansion on average. In particular some layers can be contractive, as long as the ratio of their output to the size of the input to the network is large. Further they give a simple iterative algorithm. The intuition behind it is clear, however the technical analysis is complicated by the fact that the iterates depend on the randomness. All the reviewers agreed that this paper takes a definitive step forward on an important class of problems.